# Effects of Aging on the Microstructure and Properties of 7075 Al Sheets

**DOI:** 10.3390/ma13184022

**Published:** 2020-09-10

**Authors:** Zhongxin Zhao, Ruoqing Wu, Bo Wang, Mingchu Huang, Guopeng Lei, Fenghua Luo

**Affiliations:** 1State Key Laboratory of Powder Metallurgy, Central South University, Changsha 410083, China; zhongxinzhao@csu.edu.cn (Z.Z.); guopeng129@csu.edu.cn (G.L.); 2School of Materials Science and Engineering, Central South University, Changsha 410083, China; wuruoqing@csu.edu.cn (R.W.); hmc1698@csu.edu.cn (M.H.); 3Sichuan Aerospace Changzheng Equipment Manufacturing Co. Ltd., Chengdu 600100, China; wangbo18980769957@163.com

**Keywords:** 7075 Al alloy, aging behavior, precipitation strengthening, mechanical properties

## Abstract

The effects of one-step aging and double aging on the properties and microstructures of 7075 Al sheets were studied via mechanical property testing, scanning electron microscopy, and transmission electron microscopy. The results indicated that with continued one-step aging, the tensile and yield strengths of the Al sheets first increased rapidly with an increase in the treatment time to 8 h and then increased slightly with a further increase in the treatment time to 10 h. The tensile and yield strengths became constant after 16 h of treatment. The mechanical strength properties of the Al sheets peaked after 16 h of one-step aging. However, the double aging treatment provided better mechanical properties and working efficiency than the one-step aging treatment. The tensile strength and microhardness resulting from double aging were greater than those resulting from one-step aging by 5.87% and 8.71%, respectively. Herein, we quantified the contribution ofvarious strengthening mechanisms.

## 1. Introduction

Al alloys (7075) are 7xxx series age-hardened Al alloys that are widely used in the aerospace and automotive fields owing to their low density, high strength, and moderate fatigue resistance [1,2,3]. Generally, the heat treatment of 7xxx series Al alloys includes solid solution and aging treatments, with aging treatments having a higher impact on performance. The microstructure of Al–Zn–Mg–Cu alloys changes from a solid solution to Guinier–Preston (GP) zones, then to metastable precipitate η’, and finally to the formation of the equilibrium phase η during heat treatment [4,5]. Owing to the influence of aging treatment on the type, quantity, size, and distribution of precipitated phases in the alloy matrix, it determines the final strength, plasticity, and corrosion resistance of Al alloys [6]. Therefore, aging treatment has frequently been a research focus in studies considering the properties of Al alloys [7]. While the T6 condition can result in exceptional strength, the antistress corrosion resistance and impact property are poor [8,9]. Therefore, multistep aging processes have been developed in an attempt to control the structure and performance [10]. For example, double aging (DA) treatment can significantly reduce the aging time for the T6 properties of 7075 [11]. For alloys that display the evolution of a series of metastable phases, a low temperature at the first aging state will form a high density of GP zones in the matrix. These fine precipitates will assist the nucleation of η’ phases during the second aging stage. At a high temperature with the formation of a more stable phase (such as η’ phase), it is impossible to evolve GP zones again at a lower temperature from a thermodynamic perspective [12]. The influence of the Al–Zn–Mg–Cu alloy composition on precipitation in various alloys, the evolution of mechanical performance after one-step and DA treatments, and some qualitative strengthening effects of precipitates have previously been studied [13,14]. Many different DA treatments have been researched; however, practical application parameters to improve production efficiency and save energy have not been optimized. In the present work, mechanical testing and microstructural characterization of 7075 Al sheets subjected to one-step and DA treatments are discussed. Furthermore, the contribution of each strengthening mechanism was quantified with the goal of developing a deep understanding of the one-step aging and DA treatments of 7075 Al sheets in isothermal aging.

## 2. Material and Methods

Table 1 lists the chemical composition of the 7075 Al sheets. After casting, the 7075 Al alloy was hot extruded (the extrusion temperature was 450 °C) from the Al sheets (initial thickness of 13 mm) with a reduction ratio of 8:1 to damage the as-cast microstructure; it was then subjected to a cold rolling process to obtain rolled sheets with a final thickness of 2 mm. Before the heat treatment, the alloy sheets were cut into small samples with dimensions of 195 mm × 25 mm × 2 mm. The samples were solid solutiontreated at 468 °C in molten nitrates for 20 min. They were then immediately quenched in water and cooled to room temperature. One-step aging was implemented at 140 °C for 8, 10, 12, 14, and 16 h to obtain optimal properties. DA treatment was performed as follows: the samples were maintained at 120 °C for 3 h, maintained at 160 °C for 3 h, and finally air-cooled. Figure 1 summarizes the specific process.

After heat treatment, the corresponding mechanical properties and microstructure of the 7075 Al sheets were tested and analyzed. The hardness of the samples was measured using a Shimadzu HMV-G21ST Vickers microhardness tester (Shimadzu, Kyoto, Japan); the load was HV 0.2 kgf with a keeping time of 10 s, and the average of five hardness values for each sample was uniformly taken as the final result. After grinding and polishing, the Kelle etchant (1 mL HF, 1.5 mL HCl, 2.5 mL HNO_3_, and 95 mL H_2_O) was applied to corrode the surface of the samples. Metallographic images under different magnifications were obtained using a Leica DM 2700M metallographic microscope (Leica, Wetzlar, Germany). The tensile properties of the alloys were measured using an Instron 8802 electrohydraulic servo testing instrument (the tensile rate was 2 mm/min, Instron, Norwood, MA, USA), and the fracture morphologies were observed with a Nova Nano-SEM230 scanning electron microscope (FEI company, Hillsboro, OR, USA). Transmission samples were pre-ground to 50 µm, and a 30% methanolic nitric solution was used at −25 °C. A JEOL-2000F field transmission electron microscope (TEM, JEOL, Akashima, Tokyo, Japan) was used to observe the microstructures of the alloys and the number and size of the precipitated phases at the crystal and grain boundaries. In order to investigate the effect of precipitated phase strengthening on the tensile strength of the samples and the formation of dislocations, a few TEM foils were selected from the gage length section, which was located in the deformation area of the tensile sample.

## 3. Results

### 3.1. Microstructure Analysis Results

Figure 2 shows the optical microstructure of the 7075 Al sheets under different heat treatment protocols. The morphologies were analyzed along the axis parallel to the rolling direction.

Figure 2a shows the longitudinal section microstructure of the cold-rolled sheet along the rolling direction. It can be observed that the original cold rolling microstructure exhibited very fine metal fiber flow line structures. The metallographic structures of the samples with five different hold times at 140 °C were almost similar. The images of the samples with the hold times of 8 and 16 h, which were used to analyze the subsequent results, are shown in Figure 2b,c, respectively. Figure 2d shows the metallographic structure after DA treatment. As can be seen in Figure 2b–d, the base aluminum phases were coarse grains with distinct grain boundaries. These coarse grains exhibited significant rollingdirection oriented growth. In order to better describe the grain characteristics of the oriented growth grains, the maximum length and maximum width of each grain were measured by Image Plus Pro software (version 6.0, Media Cybernetics, Rockville, MD, USA), and then the average value was obtained. For example, the average length and width of the grains after one-step aging at 140 °C for 16 h were 281.1 and 31.7 µm, respectively, and the length and width of the grains in the DA sample were approximately 237.2 and 28.6 µm, respectively.

The grain characteristics of this oriented growth are related to the recrystallization nucleation and growth mechanism of the alloy during solid solution treatment. For 7075 alloy, a large amount of MgZn_2_ (η) second-phase particles is formed when solidified and crystallized at equilibrium. These MgZn_2_ particles break during cold deformation and are aligned along the rolling direction. Figure 2a shows some fine granular compounds, most likely the MgZn_2_ phase. During the recrystallization process of the solid solution treatment, the grain boundary movement would be impeded. Because these particles will be distributed in a streamline, the resistance of grain growth in the thickness direction of the rolled plate is greater than that in the rolling direction, and the rate of recrystallized grain growth in the rolling direction is higher than in the transverse direction. Many black patches with the appearance of corrosion pits (highlighted by the short red line) can be observed in Figure 2b,c; these pits were oriented. These directional states are marked with red lines in the images. In the solidsolution process, though the MgZn_2_ phase should be completely dissolved, some of the particles do not dissolve completely owing to insufficient diffusion. Furthermore, even in the case of complete dissolution, element enrichment occurs at the original particle position. As a result, this position can be easily etched using a metallographic corrosion solution so as to show the corrosion pit. The number of corrosion pits in Figure 2c appears to be less than that in Figure 2b, which is due to the increase in aging time and the aging precipitation degree. From the perspective of thermodynamics, the energy state of the alloy decreases, thus increasing the resistance towards corrosion. From this perspective, Figure 2d has fewer corrosion pits, indicating a lower energy state and complete aging precipitation.

### 3.2. Hardness and Mechanical Properties

The tensile properties of the 7075 Al sheets after different one-step aging treatments at different times are illustrated in Figure 3. The 0h curve corresponds to the tensile properties of the sample subjected to the solid solution treatment at 468 °C for 20 min.

Figure 3a shows that the tensile strength of the samples first increased rapidly with an increase in the treatment time up to 8 h and then increased slightly with a further increase in the treatment time to 10 h. The tensile strength became constant after 10 h of treatment time. The trend shown by the yield strength of the samples was consistent with that shown by the tensile strength and opposite to that shown by the elongation. Figure 3b shows the magnified mechanical property curves of the samples over the aging treatment time range of 8–16 h. As can be observed from Figure 3b, the tensile strength first increased from 503.2 MPa to the peak value of 515 MPa with an increase in the aging time from 8 to 10 h and then decreased slightly to 511.6 MPa with a further increase in the aging time up to 16 h.

In general, the aging sequence of Al–Zn–Mg–Cu alloys isthe GP zones followed by the η’ phase. At the initial stage of aging, the GP zones precipitate first, and the enhancement effect of the η’ phase is more significant at the later stage. Therefore, it can be inferred that the strength peak in the early aging period of 10 h came from the strengthening of the GP zones. The aging time affected the coarsening and dissolution of the GP zones, η’ phase fraction, and the source of the strength peak. It can be seen from Figure 3a that the change in the tensile property caused by aging time was not exceedingly significant. However, Figure 3b shows that the elongation rate was also high when the strengthening peak appeared at the beginning of aging, which is a significantphenomenon and requires further exploration.

Table 2 lists the values of the tensile and microhardness properties of the 7075 Al sheets at different heat treatment states.

Ascan be observed in Table 2, in the one-step aging process and with an increase in the aging time, the hardness first increased, then decreased, reaching 183.6 HV after 16 h. However, with the DA treatment, the hardness obtained was higher and reached 199.6 HV. Overall, one-step aging at 140 °C for 16 h gave the best performance among the one-step processes. As expected, the DA treatment resulted in a significant increase in both the hardness and strength. The elongation after the DA treatment decreased only slightly compared to that after one-step aging.

### 3.3. Morphology of Tensile Fracture

Figure 4 shows the tensile fracture morphology of the 7075 Al sheets after different aging conditions. Figure 4a–e shows the fracture surfaces of the samples after undergoing a one-step aging treatment for 8, 10, 12, 14, and 16 h, respectively. Figure 4f shows the same image for the sample subjected to a DA treatment.

The samples showed mixed ruptures with transgranular fractures and intergranular fractures. A number of tearing edges (indicated by red arrows), some small secondary particles, a number of fine dimples around the fractures (marked by the black circle A), and intergranular fractures (indicated by the black circle B), could be observed. Therefore, the fractures following the aging processes can be attributed to ductile failure. In addition, the fracture morphology of the samples depended on their aging conditions. As can be observed in Figure 4a, the sample aged for 8 h showed a large number of shear zones and dimples. From Figure 4a–e, it can be observed that with an increase in the one-step aging time from 8 to 16 h, the density of the dimples and shear zones decreased, and the proportion of transgranular fractures also decreased significantly, while the intergranular fraction increased. This indicates that the fracture mode changed from transgranular dimple and transgranular shearing mixed fractures to intergranular fractures with an increase in the one-step aging time. As shown in Figure 4e,f, the DA treatment produced fewer large secondary phase particles compared to the 16 h one-step aging treatment. Moreover, the plasticity of the samples also decreased after the DA treatment [13]. The DA-treated samples showed fewer tearing edges and shear zones and larger dimples than the 16 h one-step aged sample. This indicated that the DA-treated 7075 Al sheets showed mainly intergranular fractures like the one-step aged (16 h) sheets. However, the proportion of intergranular fractures was only slightly higher than that in the case of the one-step aged sheets.

### 3.4. TEM Analysis

Aging is an important heattreatment process used to improve the properties of Al alloys, and the type and distribution of the precipitated phases affect the performance of the material.

The samples with ideal mechanical performance were divided into two groups for further investigation: those subjected to one-step aging for 16 h and those subjected to the DA treatment. Figure 5a,b shows the bright field (BF) TEM images and the corresponding selected area diffraction patterns (SADPs) of the one-step and the DA-treated samples along the [001]_Al_ zone axis. In the [001]_Al_ zone axis, two types of η’ structures, namely, plate-like and round-shaped, were identified.

As shown in Figure 5a,b, the precipitation density of the DA treatment sample was greater than that of the one-step aged sample; however, the sizes of the precipitates in the two cases were similar. An analysis of the diffraction images identified diffraction spots for the η’ phase and the GP zones, where the GP zone diffraction spots are located at 1/3 {422} and the diffraction spots of the η’ phase are located at 1/3 and 2/3 of the {220} plane [13]. Figure 5a,b shows that the second-phase diffraction spots exhibited after DA treatment were brighter than those after one-step aging, while the diffraction spots in the GP zone were dimmer, indicating an increase in the number of precipitated phases and a decrease in the number of GP zones formed in the early stages of aging. The fine and uniformly distributed η’ precipitates in the matrix are in a semicoherent relationship with the matrix and can hinder the movement of dislocations, thus increasing the strength of the alloys [15].

In the 7075 Al sheets, TEM foils were selected from the gage deformed area of the tensile sample to analyze the strengthening of the dislocations. Tangles or networks of dislocations were formed after the aging process (indicated by red arrows in Figure 6a,b). Moreover, the dislocation density of the DA treatment sample was higher than that of the one-step aged sample.

With the parameters of the one-step and DA treatment grains (as shown in Figure 2 and Section 3.1), the strengthening effect of the grains size was found to be less than 10 MPa. The calculation was carried out according to Equation (4), as discussed further on. However, under experimental conditions, the strength of the sample was high, indicating that the subgrains played an important role in the strengthening mechanism. The average length and width of the subgrains after one-step aging were 1.2 and 0.5 µm, respectively. The length and width of the elongated subgrains in the DA treatment sample were ~1.0 and ~0.4 µm, respectively, according to the standard [16]. However, it is essential to explore the characteristics of precipitation, precipitation size ranges, average size of the precipitate, etc. Thus, in view of the limited methodology available, the quantitative details of the precipitates were gathered with Image Plus Pro (IPP). Considering the structure of the precipitates, the length of the long axis could be treated as the diameter, and the sizes of the elliptical precipitations were treated in the same manner. Figure 7 presents the distribution of different diameters of η’ (the uncertainty in the diameter is 1 nm), and it should be noted that extremely small or large particles were ignored.

According to statistical analysis, the size of the precipitates was concentrated within the range of 7–11 nm for both the one-step and two-step aging processes (shown in Figure 7a). To minimize the error, a 95% confidence level was set according to Equation (1):(1)$$P\{D_{1}<D_{x}<D_{2}\}=1-a$$
where *P* represents the probability that some precipitate diameter, *D_x_,* will be between *D*_1_(min) and *D*_2_(max) and it amounts to 95% (with 5% probability of a type I error, i.e., 100 − 5 = 95%) with a value of *D*_1_ and *D*_2_ of 5 and 14 nm, respectively. The average size of the precipitates was calculated from Figure 7b,c with Equation (2):(2)$$D_{mean}=\frac{1}{n}\sum_{x=1}^{n}D_{x}f_{x}$$
where *f_x_* is the corresponding fraction percent of the precipitates. The resulting average diameter of the precipitates after one-step aging was 9.05 nm. Applying the same method, the average precipitate size after two-step aging was 8.72 nm. The method of convergent beam diffraction can be used to confirm the sample thickness [17]. To reduce the error of convergence, the reliable thickness should exceed 50 nm; however, considering the high density of precipitates, the thickness should be less than this value [17]. Thus, the maximum sample thickness of 50 nm was adopted for one-step and two-step aging processes in the following analyses. On the basis of some helpful hypotheses proposed by other researchers [18], the precipitate volume of the one-step aged samples was calculated using Equation (3):(3)$$V_{p}=V_{sphere}+2V_{platelet}-V_{cor}$$
where *V_sphere_* is the volume of observed round particles and *V_sphere_ = 4/3 πR_s_^3^*, *V_platelet_* is the volume of the observed platelets and *V_plate_*
*= π/4TD^2^* (*T* is the thickness of the precipitate, and *T* ≈ 0.38 *D*) [18], and *V_cor_* is the volume of the platelets with round precipitates and *V_cor_ = 4/3 πR_mean_^3^*. The volume fraction is *φ = V_p_/V*, where *V* represented the volume of the sample, which was replaced by the area of the TEM images multiplied by the sample thickness. The result of the volume fraction of the precipitations was 5.25 ± 0.3% and 6.57 ± 0.3% for the one-step aging at 140 °C for 16 h and two-step aging treatments, respectively.

## 4. Discussion

In an Al alloy, including the extruded Al 7xxx series alloys, several types of strengthening mechanisms are at play, including solid solution strengthening, precipitation strengthening, dislocation strengthening, and grain boundary strengthening [19]. In this work, different aging conditions were studied and the Hall–Petch Equation (4) was used to analyze the grain boundaries [20]:(4)$$\Delta \sigma_{HP}=\sigma_{f}+k_{i}d^{-m}$$

Here, Δ*σ_HP_* is the increment in yield strength, *σ_f_* is the yield strength for pure Al at the initial grain state (approximate value 16 MPa [21]), *k_i_* is the Hall–Petch constant (0.065 MPa·*m*^−½^ for pure aluminum) [22]), *d* is the typical average subgrain size (typically 1.39 and 1.08 µm), and *m* is an index for grain boundary strengthening (typical value of 1/2). Table 3 lists the results obtained using the corresponding equations.

Dislocation strengthening follows Taylor’s relationship as described in Equation (5):(5)$$\Delta \sigma_{d}=\frac{BGb}{L}$$

Here, *B* is 0.2 for FCC metals, *G* is the shear modulus and is 26.9 GPa for Al–Zn–Mg–Cu alloys, *b* is the Burgers vector and is 0.284 nm, and *L* is the average interparticle spacing as described in Equation (6) [23]:(6)$$L=r_{p}\{{\left(2\pi /3f\right)}^{1/2}-{\left(8/3\right)}^{1/2}\}$$

In Equation (6), *r_p_* represents the radius of η’ with values (4.53 and 4.36 nm) as discussed in Section 3.4, and *f* is 0.0525 and 0.0657 for the one-step aging and DA treatment processes, respectively. Thus, for one-step aging, *L* = 4.53 × {(2× 3.14/3 × 0.0525)^1/2^ − (8/3)^1/2^}and its value was approximately 21.21 nm. Therefore, the expression of Δ*σ_d_* is (0.2 × 26.9 × 10^3^ × 0.284)/21.21 and its value was approximately 71.1 MPa. A similar calculation procedure was applied to the dislocation strengthening of double aging and the following strengthening mechanisms, and it did not list specific expressions.

Precipitation strengthening can be calculated using Equation (7) [24]:(7)$$\Delta \sigma_{ppt}=A\frac{1}{\delta \sqrt{3}(\sqrt{\pi /6f}-\sqrt{2/3})}\ln\frac{\sqrt{2}\delta }{r_{0}}$$
where *A* is a constant equaling 0.85*Gb*/2*π(1−ν)*^1/2^*,* ν represents the Poisson ratio value of 0.3 for Al alloys, *δ* is the diameter for the round precipitates (with values 9.05 and 8.72 nm as discussed in Section 3.4), and *r*_0_ represents the core radius of the dislocation (ca. 0.6 nm) [25].

Solid solution strengthening also plays a part in the overall strengthening process. However, as some Mg and Zn species form precipitates in a solid solution, the contribution of the solid solution atoms will be less than that under ideal conditions. Therefore, we did not consider solid solution strengthening here.

The total yield strength results from the different mechanisms can be summarized as follows [26]:(8)σtot=ΔσHP+M{(Δσd2+Δσppt2)12}
where *M* is the Taylor factor equal to 3 [27]. The total yield strength values of the samples are listed in Table 3.

The experimental yield strengths (Table 2) of the 16 h one-step aged and DA-treated samples were higher than their calculated yield strengths (45.9 and 22.9 MPa, respectively). This may result from lattice strengthening, solid solution strengthening, modulus strengthening, and/or stacking fault strengthening. Moreover, it is possible that precipitation strength plays a more significant role than dislocation interaction because the formation of precipitate leads to an increase in the number of subgrains and hinders further plastic deformation.

It is interesting that this work is similar to a study by S.V. Emani [11], which involved DA treatment of 7075 extrusions. In that paper, aging at 121 °C for 55 min and then aging at 177 °C for 55 min resulted in a tensile strength of 548 MPa, yield strength of 417 MPa, elongation of 16%, and microhardness less than 191 HV. In this study, the rolling sheets showed a tensile strength of 541.4 MPa, a yield strength of 491.1 MPa, an elongation of 10.7%, and a microhardness of 199.6 HV after DA treatment. The tensile strength of these sheets was slightly lower than that of the 7075 extrusions because of the extrusion effect. The aging process used in this study was for a longer duration; however, the aging temperature was lower than those reported previously. Owing to the fact that DA treatment parameters have not been optimized and widely accepted, the new double aging process was implemented along with different one-step aging treatments, and the results were studied and compared. In summary, it should be noted that the DA treatment applied in this study markedly reduced the procedure from 16 to 6 h. The microhardness and tensile strength both increased beyond that of conventional one-step aging, and the ductility dropped slightly. This method can save energy and improve the production efficiency in practical industrial production. The role of various strengthening mechanisms on the mechanical behavior was also analyzed and the results summarized.

## 5. Conclusions

In this work, we studied different aging treatments for 7075 Al sheets. By studying the microstructures, we gained insight into several strengthening mechanisms. The mechanical performance and microstructure details can be summarized as follows:During one-step aging, the strength and hardness of the 7075 Al sheet first increased, then decreased. The tensile properties peaked after aging for 10 h; the tensile strength and microhardness were 515.3 MPa and 179.8 HV, respectively. Double aging of the Al sheet gave better mechanical properties than one-step aging; the tensile strength and microhardness were increased by 33.5 MPa and 12.2HV, respectively.The microstructural analysis showed that the samples subjected to double aging had higher dislocation and precipitation densities. Double aging resulted in the formation of a large number of distributed η’ phases and GP zones in the matrix within a short period of time. The strength and working efficiency of the sheets improved significantly.Precipitation had a significant influence on the 7075 Al sheets compared to dislocation.

## Figures and Tables

**Figure 1 materials-13-04022-f001:**
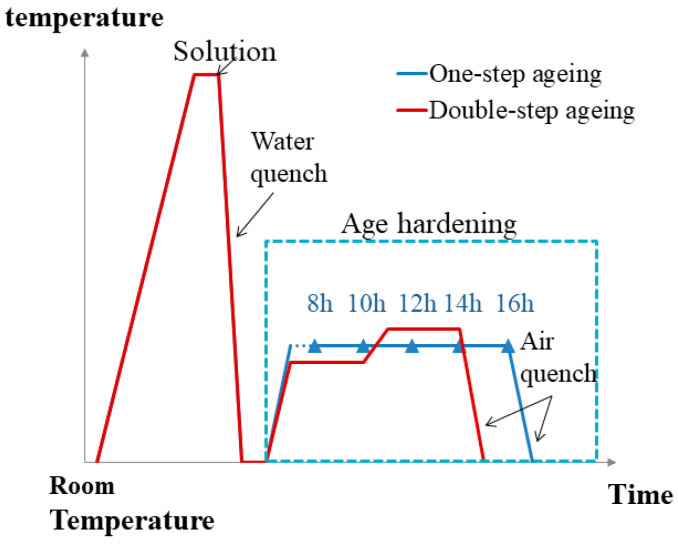
Schematic of the heat treatment process for 7075 Al sheets. Blue curves indicate the one-step aging process at 140 °C for 8, 10, 12, 14, and 16 h. Red curve denotes the double aging (DA) treatment at 120 °C for 3 h, followed by heating at 160 °C for 3 h.

**Figure 2 materials-13-04022-f002:**
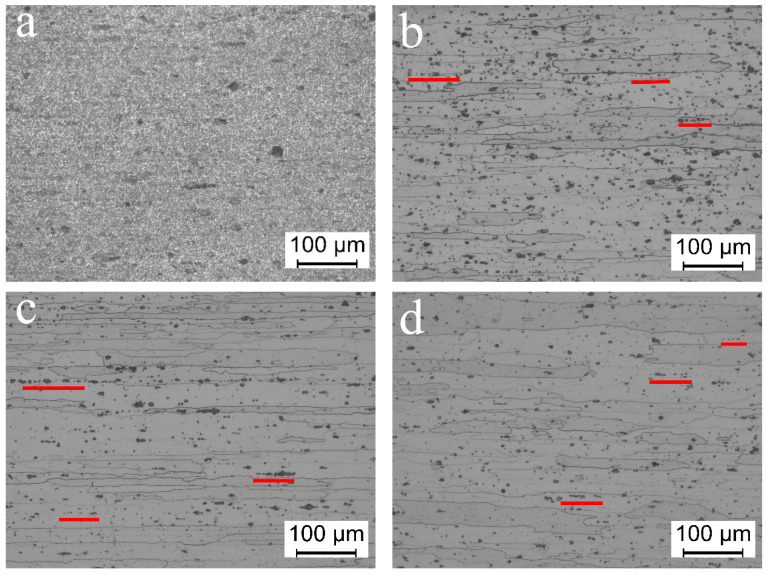
Longitudinal section optical microstructures of the 7075 Al sheets subjected to an aging temperature of 140 °C for different durations: (**a**) cold rolling state, (**b**) 8 h, (**c**) 16 h one-step aging, and (**d**) DA (3 h at 120 °C and 3 h at 160 °C) treatments; horizontal line represents the rolling direction.

**Figure 3 materials-13-04022-f003:**
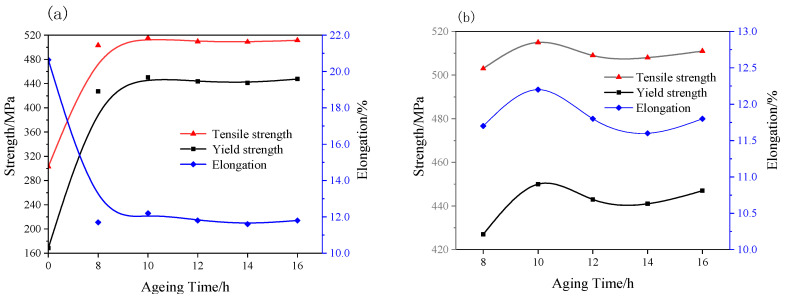
Tensile properties of the 7075 Al sheets subjected to different one-step aging treatments. (**a**) The tensile properties during the whole process and (**b**) a section of the process.

**Figure 4 materials-13-04022-f004:**
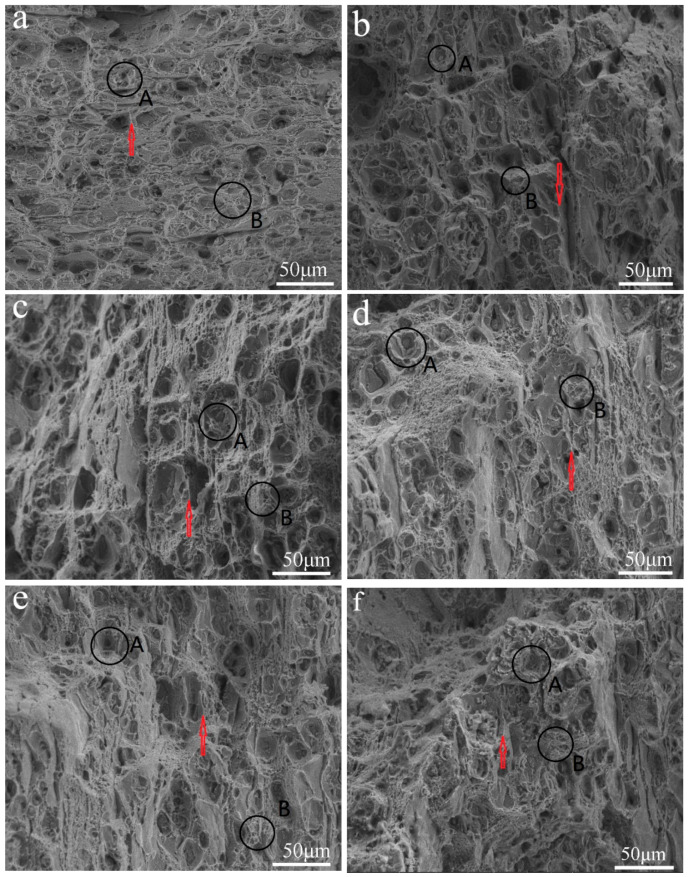
Tensile fracture morphologies of the 7075 Al sheets subjected to one-step aging for various durations: (**a**) 8 h, (**b**) 10 h, (**c**) 12 h, (**d**) 14 h, and (**e**) 16 h at 140 °C; (**f**) DA treatment (3 h at 120 °C and 3 h at 160 °C).

**Figure 5 materials-13-04022-f005:**
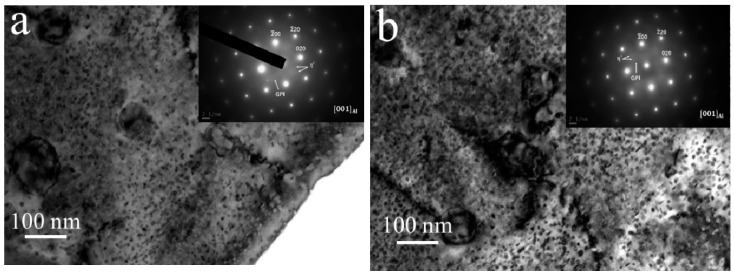
Bright field (BF) TEM images and diffraction patterns of the 7075 Al sheets in the [001]_Al_ zone axis: (**a**) one-step aging for 16 h at 140 °C and (**b**) DA treatment (3 h at 120 °C and 3 h at 160 °C).

**Figure 6 materials-13-04022-f006:**
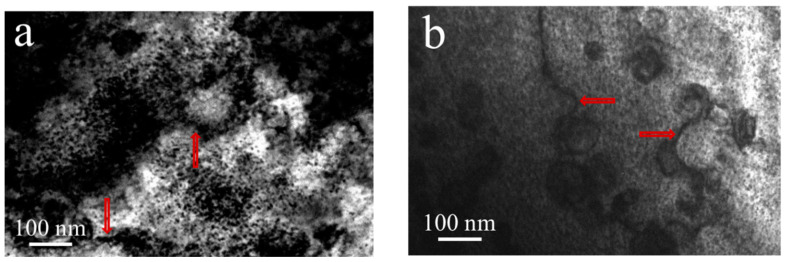
Microstructures of the dislocations of the 7075 Al sheets along the [001]_Al_ zone axis: (**a**) one-step aging for 16 h at 140 °C and (**b**) DA treatment (3 h at 120 °C and 3 h at 160 °C).

**Figure 7 materials-13-04022-f007:**
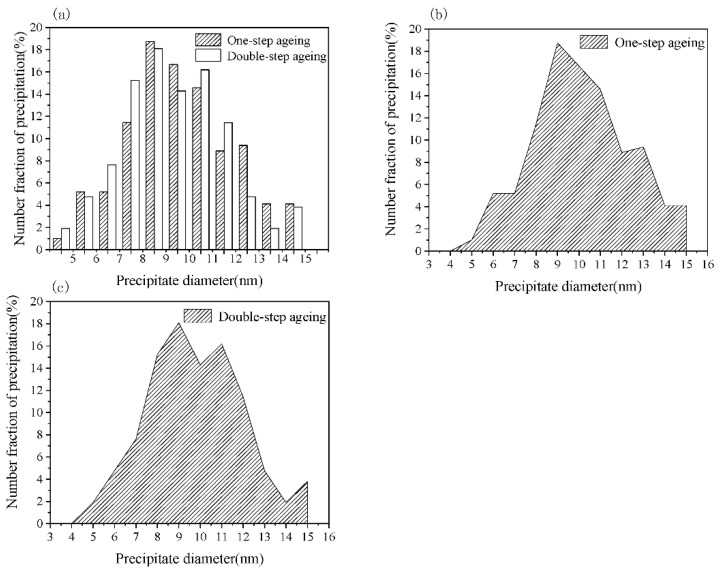
Precipitate size distributions: (**a**) histogram (the uncertainty is 1 nm) and the shadow region diagram of (**b**) one-step aging for 16 h at 140 °C and (**c**) DA treatment (3 h at 120 °C and 3 h at 160 °C).

**Table 1 materials-13-04022-t001:** Chemical composition of the 7075 Al alloy (mass fraction, %).

Alloy	Zn	Mg	Cu	Mn	Fe	Si	Cr	Ti	Al
7075	5.72	2.55	1.50	0.13	0.15	0.20	0.23	0.12	Bal.

**Table 2 materials-13-04022-t002:** Mechanical properties of the 7075 Al sheets subjected to different aging treatments.

Aging Time	YS (MPa)	UTS (MPa)	Elongation (%)	Microhardness (HV)
0 h	168.7	302.9	20.6	36.4
8 h	427.1	503.2	11.7	173.4
10 h	450.3	515.3	12.2	179.8
12 h	443.6	509.5	11.8	168.8
14 h	441.2	508.8	11.6	170.4
16 h	447.5	511.6	11.8	183.6
DA treatment	491.1	541.4	10.7	199.6

YS is yield strength and UTS is ultimate tensile strength.

**Table 3 materials-13-04022-t003:** Strengthening increment by different strengthening mechanisms.

Strengthening Increment, MPa	One-Step Aging (16 h)	DA Treatment (3 h at 120 °C and 3 h at 160 °C)
Grain boundary strengthening, Δ*σ_HP_*	71.1	78.6
Dislocation strengthening, Δ*σ_d_*	72.0	87.3
Precipitation strengthening, Δ*σ_ppt_*	83.4	96.1
Yield strengthening Increment, Δ*σ_tot_*	401.6	468.2

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
