# Peer review of "Effects of Aging on the Microstructure and Properties of 7075 Al Sheets"

_materials, 2020, doi:10.3390/ma13184022_

Round 1
Reviewer 1 Report
JMSC-D-19-07306 R1
In the revised version the authors have clarified a few topics yet, as detailed below, I still believe that the paper – both the experimental work and the theoretically analysis of the results – has too many inconsistencies to be publishable.
- There is not enough information on the starting material (possible homogenization, extrusion temperature, extruded thickness, …). Also the chemical composition of the material at hand is not given; Table 1 merely lists the registered composition range for alloy AA 7075.
- In Sec. 3.1 the authors explain the formation of recrystallized grains during the solution heat treatment. They also address the occurrence of MgZn2 particles. Since they (correctly) assume that MgZn2 particles would dissolve during the solution heat treatment (otherwise the material would not harden upon subsequent ageing), they fabulate about corrosion pits. Again, I am not really an expert in AA 7xxx series alloys, but in my opinion, the micrographs simply show Fe-bearing constituent phases (provided the alloy does contain Fe – see above), which usually do not dissolve during the solution heat treatment.
- In Sec. 3.2 the authors stress the double strength peak after ageing for 10 h and 16 h at 140°C. Although the explanation (first formation of GP-zones and subsequent formation of h’-particles) may be correct, it is not supported by the experimental results. Unfortunately, TEM micrographs for 10 h are not shown, and the quality of the ones presented in Fig. 5 is not good enough to support the findings (at least in the PDF document I have available). By the way, the differences in yield strength between 10 h, 12 h, 14 h and 16 h are small, and the hardness results show a different behavior (maximum hardness at 16 h as opposed to max. yield strength after 10 h). Thus, there is a question mark over accuracy and reproducibility of the mechanical data.
- To me, Sec. 3.3 on fracture morphology is largely irrelevant and does not really add any important information to support the findings of the present study.
- In Sec. 3.4 the authors describe very accurately how the average particle sizes were measured. However, they do not explain how the volume fraction of particles was determined. Information on the maximum valid(?) thickness of TEM samples is not helpful, but the actual TEM foil thickness is required. The explanation of Vcor in Eq. 2 is inconclusive.
- In Sec. 4 the authors attempt at using the microstructure data to estimate the resulting strength. In the assessment of the Hall-Petch contribution the subgrain sizes are used, which is meaningful. However, the authors do not describe how the subgrain sizes were determined. The numbers given in Sec. 3.4 and Sec. 4 differ. The contribution of dislocations is derived from the particle statistics, which is probably ok. However, analysis of the yield strength as opposed to the ultimate tensile strength would not require dislocations, which would make the comparison simpler. The contribution of hardening phases is surprisingly small for peak-aged 7075, which may be due to the issue of volume fraction and TEM foil thickness addressed above. In general, the authors use symbol s interchangeably for tensile strength (s) and shear strength (t). These quantities should be differentiated properly.
- In the revised version the discrepancy between simulated strength and experimental strength is huge, which puts the entire exercise into question. Unfortunately, the authors do not critically scrutinize this discrepancy, but merely list a few points – lattice strengthening, solid solution strengthening, modulus strengthening, and/or stacking fault strengthening – which, in their opinion, may be responsible for this discrepancy. This is definitely not sufficient for a scientific publication!
=============================================================================
JMSC-D-19-07306
The manuscript JMSC-D-19-07306 "Effects of ageing on the microstructure and properties of 7075 Al sheets" has been reviewed. In this reviewer’s opinion, this paper has too many inconsistencies (see below) and hence should not be published, at least in its present state.
- The material has been solution-heat treated for 20 min at 468°C before age hardening. This appears to be a rather conventional heat treatment. I am not certain whether or not 7xxx series alloys recrystallize during solutionizing (6xxx series alloys usually do), but at least they should recover heavily. So I am surprised that the authors find so high dislocation densities.
- Which particles do we see in Fig. 2? Constituent phases and dispersoids should not change during ageing at 140 or 160°C! Likewise, the grain structure should not change as well.
- The authors state that “… with continued ageing, the tensile strength and hardness of the Al sheets increase first, then decrease, reach maxima, and finally decrease.” To me this is rather unusual, but the authors do not provide an explanation for this phenomenon. It would be helpful to show the complete hardening curves, i.e. add the for 0 h of ageing.
- There is no information on how the volume fraction of particles was measured. Volume fractions of 0.12 and 0.16 (i.e. 12 or 16%?) appear very high. Did the authors take the TEM foil thickness into consideration?
- It is very interesting to use the microstructure data to estimate the resulting strength. However, to me there are a number of questionable values in Sec. 4 and Table 3 which need to be addressed more thoroughly. Also, the choice of the hardening equations (Eq. (5) and (6)) appears somewhat arbitrary.
Author Response
We thank the reviewer for patient and constructive comments and our respond to specific points as below.
Comment 1.There is not enough information on the starting material(possible homogenization, extrusion temperature, extrudedthickness, ...). Also the chemical composition of the material athand is not given; Table 1 merely lists the registered compositionrange for alloy AA 7075.Reviewerpointed out that the information on the starting materialwas absent and the chemical composition of the material athand was not given.
Reply: Thanks for your kind comment. we have realized those problems and have added the corresponding extrusion parameters in our paper. Besides, we adopted a SPECTRO MAXx direct-reading spectrometer to verify the special chemical compositions and have directly modified those compositions’ mass fraction in our paper.
Comment 2.In Sec. 3.1 the authors explain the formation of recrystallizedgrains during the solution heat treatment. They also address theoccurrence of MgZn2 particles. Since they (correctly) assumethat MgZn2 particles would dissolve during the solution heattreatment (otherwise the material would not harden uponsubsequent ageing), they fabulate about corrosion pits. Again, Iam not really an expert in AA 7xxx series alloys, but in myopinion, the micrographs simply show Fe-bearing constituentphases (provided the alloy does contain Fe - see above), Whichusually do not dissolve during the solution heat treatment.
Reply: According to the Fig. 2, Fe-bearing constituent phases may usually do not dissolve during the solution heat treatment.But, some important specialities couldbe observed, such as the average length and width of grain as well as the features of small black patches. There were not equiaxed grains formed or fully recrystallization occurred, indicated that the parameters of solution treatment were proper.
Comment 3.In Sec. 3.2 the authors stress the double strength peak after ageing for 10 h and 16 h at 140°C. Although the explanation (first formation of GP-zones and subsequent formation of h’-particles) may be correct, it is not supported by the experimental results. Unfortunately, TEM micrographs for 10 h are not shown, and the quality of the ones presented in Fig. 5 is not good enough to support the findings (at least in the PDF document I have available). By the way, the differences in yield strength between 10 h, 12 h, 14 h and 16 h are small, and the hardness results show a different behavior (maximum hardness at 16 h as opposed to max. yield strength after 10 h). Thus, there is a question mark over accuracy and reproducibility of the mechanical data.
Reply:Firstly, for the problem of ‘TEM micrographs for 10 hare not shown’, due to our main purpose was to compare the difference between one-step peak aging and double aging and may not pay special attention to the microstructure of sample of ageing for 10 h. There’s no denying that some more experimental results needed for supporting the double strength peak. About this interesting point, we will do further research. Secondly, for the quality of TEM micrographs, we have replaced the micro markers and have did some small adjustments to improve the quantity of those pictures. Thirdly, the reviewer questioned the mechanical data. For the finding that the variation of microhardness was inconsistent with the change of yield strength, the data was accuracy firstly. It’s may need more microstructural analyses and we will do some work in further scientific research.
Comment 4.To me, Sec. 3.3 on fracture morphology is largely irrelevant and does not really add any important information to support the findings of the present study.
Reply: Thanks for your kind comment. Our view was that the morphology of tensile fracture was the important evidence to verify the fracture mechanism for materials under different ageing conditions. In the revised version, we have made a more detailed analysis of the fractures, trying to make the analysis more valuable and persuasive.
Comment 5.In Sec. 3.4 the authors describe very accurately how the average particle sizes were measured. However, they do not explain how the volume fraction of particles was determined. Information on the maximum valid(?) thickness of TEM samples is not helpful, but the actual TEM foil thickness is required. The explanation of Vcor in Eq. 2 is inconclusive.
Reply: Thanks for your kind comment.we have added some details about the source of the measurement of particles’ volume fraction in our study. And we have described the actual TEM foil thickness in detail. Please see in revision manuscript.
Comment 6.In Sec. 4 the authors attempt at using the microstructure data to estimate the resulting strength. In the assessment of the Hall-Petch contribution the subgrain sizes are used, which is meaningful. However, the authors do not describe how the subgrain sizes were determined. The numbers given in Sec. 3.4 and Sec. 4 differ. The contribution of dislocations is derived from the particle statistics, which is probably ok. However, analysis of the yield strength as opposed to the ultimate tensile strength would not require dislocations, which would make the comparison simpler. The contribution of hardening phases is surprisingly small for peak-aged 7075, which may be due to the issue of volume fraction and TEM foil thickness addressed above. In general, the authors use symbol s interchangeably for tensile strength (s) and shear strength (t). These quantities should be differentiated properly.
Reply:Thanks for your kind comment.Themeasurement of sub-grain sizes were applied by the standard of ASTM E 112-13, and we also added the explanation in revised manuscript. We realized the value of the contribution ofprecipitates was smaller than other strengthening mechanisms and did some further verification and correction, which could be seen in Table 3.
Comment 7.In the revised version the discrepancy between simulated strength and experimental strength is huge, which puts the entire exercise into question. Unfortunately, the authors do not critically scrutinize this discrepancy, but merely list a few points – lattice strengthening, solid solution strengthening, modulus
Reply:Generally speaking, there were didexist some shortcomings in our work, the reviewers put forward many constructive opinions on our paper and we have made some revises correspondingly.
We hope the reviewers will agree with our explanations and revisions and accept this article! If there have some points needed to further modify, please point out and we will make detailed verifications.
Thank you for your detailed reviews again!
Reviewer 2 Report
Thank you very much for your interesting study. I would like to draw your attention to some comments which are listed below:
- In Abstract: the tensile and yield strengths became constant after 10 hours according to Figure 3, not after 16 hours as it is unknown what relationship will be established after 16 hours.
- In Abstract: it is stated that the mechanical strength properties of the Al sheets peaked after 16 h. However, as can be seen from Table 2, the highest level of the properties lied at 10 h.
- The introduction can include more information about what influnce has the DA treatment on the microstructure and properties of the 7xxx series alloys.
- In the Materials and methods: what kind of etchant was used for the metallographic investigation?
- In order to investigate the contribution of the precipitated phases and dislocations, the samples were taken from the gage area of the tensile samples. What is the reason why the samples after tensile test were consider instead of the samples before test? Is it possible that the deformation affected the dislocation density e.g. precipitation density?
- In the description to Figure 2 please correct (d) DA (please, shift one-step ageing before (d)).
- What is the statement that the η´ precipitates are semi-coherent based on?
- The size of the sub-grains are mentioned. How was that measured?
- How many pictures were analyzed in order to measure the size and fraction of the precipitates? In general, how did this procedure look like?
- Please, improve the presentation of the scale bars in Figure 5.
Author Response
Response to comments ofReviewer #2
We thank the reviewer for theencouragementand constructive comments and our respond to specific points as below.
Comment 1. In Abstract: the tensile and yield strengths became constant after 10 hours according to Figure 3, not after 16 hours as it is unknown what relationship will be established after 16 hours.
Reply: Thanks for your careful comment.If the ageing time exceeded to 16 h, the time cost will be too high. On the other hand, according our previous research, the ageing stage will turn to over-ageing. So, we didn’t study the mechanical behavior when the ageing time over 16 h.
Comment 2.In Abstract: it is stated that the mechanical strength properties of the Al sheets peaked after 16 h. However, as can be seen from Table 2, the highest level of the properties lied at 10 h.
Reply: Thanks for your careful comment. According to the Table 2, The highest level of the tensile properties did lie at 10 h. But for the microhardness, it increased slightly with the ageing time up to 16 h. Considering these two indicators, we defined the ageing time of 10 h as the peak ageing treatment of tensile properties. As the variation of microhardness was inconsistent with the change of tensile strength, the data was accuracy firstly. It’s may need more microstructural analyses and we will do some work in further scientific research.
Comment 3.The introduction can include more information about what influnce has the DA treatment on the microstructure and properties of the 7xxx series alloys.
Reply:It is necessary to add some more information about influences of DA treatment on the microstructure and properties of the 7 series alloys. So, we have made some supplements in Introduction.
Comment 4.In the Materials and methods: what kind of etchant was used for the metallographic investigation?
Reply: About the etchant, we have added the information of etchant in revised manuscript, seen in Part 2. Material and methods.
Comment 5.In order to investigate the contribution of the precipitated phases and dislocations, the samples were taken from the gage area of the tensile samples. What is the reason why the samples after tensile test were consider instead of the samples before test? Is it possible that the deformation affected the dislocation density e.g. precipitation density?
Reply: In order to analyze the function of precipitate strengthening in the strengthening mechanism more directly, we directly selected the gage area of samples that undergone tensile test for microstructure analysis. We hypothesized that the density of precipitates and dislocation may be not affected by deformation. This may need further study in our following research.
.
Comment 6.In the description to Figure 2 please correct (d) DA (please, shift one-step ageing before (d)).
Reply:We have shifted ‘one-step ageing’ before(d)
Comment 7.What is the statement that the η´ precipitates are semi-coherent based on?
Reply: Considering many papers of previous scholars have identified η' phase shows a semi-coherent characteristic with matrix, we directly expressed in our paper.
Comment 8.The size of the sub-grains are mentioned. How was that measured?
Reply:Themeasurement of sub-grain sizes were applied by the standard of ASTM E 112-13, and we also added the explanation in revised manuscript.
Comment 9.How many pictures were analyzed in order to measure the size and fraction of the precipitates? In general, how did this procedure look like?
Reply:We analyzed two pictures of each samples and hundreds precipitates of each picture were counted.This process required lots of effort and time, but it was worthwhile to decrease the experimental error.
Comment 10.Please, improve the presentation of the scale bars in Figure 5.
Reply:We haveimproved this presentation accordingly.
Generally speaking, there were didexist some inadequacies in our work, the reviewers put forward many constructive opinions on our paper and we have made some revises correspondingly. We hope the reviewers will agree with our explanations and revisions and accept this article! If there have some points needed to further modify, please point out and we will make detailed verifications.
Thank you for your detailed reviews again!
Reviewer 3 Report
Manuscript ID: materials-914253
Title: Effects of ageing on the microstructure and properties of 7075 Al sheets
Authors: Zhongxin Zhao et al.
Introduction. This chapter is very small. Authors must add more information about ageing of 7075 Al alloys or describe references [1-13] in details. The novelty of the work should be written more clearly. What is the difference between Authours research and previous studies? This should be summarized in 4-5 sentences.
Materials and methods.
Table 1. The chemical composition of the alloy is represented by wide ranges of metals, why are such ranges made? Did the Authors show standard content of impurities or the composition of the sample? Then there must be an exact chemical analysis. Authors can add the value of the standard and on the second line add the exact composition of the alloy sample to show that it corresponds to the requirements.
Figure 1. Why did these modes of one-stage and two-stage ageing were selected? Why did these temperatures and holding times were chosen?
Include one SEM-figure to show how the Image Plus Pro software calculate grains. The article text mentions this program twice, but the principle of its operation is unclear.
Results
Figure 4. This SEM-images are very similar, I do not see any significant difference between these figures. Authors should show the differences more clearly.
Line 203-205. This is a repetition of already written information, delete this sentence.
I think this article is better submitted to Metals.
Author Response
We thank the reviewers for patient and constructive comments and our respond to the specific points as below.
Comment 1.Introduction. This chapter is very small. Authors must add more information about ageing of 7075 Al alloys or describe references [1-13] in details. The novelty of the work should be written more clearly. What is the difference between Authours research and previous studies? This should be summarized in 4-5 sentences.
Reply:Thanks for your careful comment. We have made some additions about details of double ageing to the introduction of our work and tried tomake the novelty more clearly.
Comment 2.Table 1. The chemical composition of the alloy is represented by wide ranges of metals, why are such ranges made? Did the Authors show standard content of impurities or the composition of the sample? Then there must be an exact chemical analysis. Authors can add the value of the standard and on the second line add the exact composition of the alloy sample to show that it corresponds to the requirements.
Reply: We have realized those problems and have adopted a SPECTRO MAXx direct-reading spectrometer to verify the special chemical compositions and have directly modified those compositions’ mass fraction in our paper.
Comment 3.Figure 1. Why did these modes of one-stage and two-stage ageing were selected? Why did these temperatures and holding times were chosen?
Reply: For the one-step ageing treatment, the mode including the ageing temperature and holding time were designed according our previous research. For the double ageing process, considering the parameters have not been optimized (seen in Section 1. Introduction) and our completed project, we designed this ageing treatment process.
Comment 4.Include one SEM-figure to show how the Image Plus Pro software calculate grains. The article text mentions this program twice, but the principle of its operation is unclear.
Reply: Themeasurement of grain sizes were applied by the standard of ASTM E 112-13, and we also added the explanation in revised manuscript.
Comment 5.Figure 4. This SEM-images are very similar, I do not see any significant difference between these figures. Authors should show the differences more clearly.
Reply:The samples showed mixed rupture with transgranular fracture and intergranular fracture and may be seemed similar. But in detail, it did exist difference. Tearing edges (indicated by red arrows), a number of fine dimples around the fracture (marked by black circle A), and intergranular fracture indicated by black circle B could be observed in SEM imagines and descriptions.
Comment 6.Line 203-205. This is a repetition of already written information, delete this sentence.
Reply: This sentence may have described in similar way. But it had its function in describing the strengthening effect. So, we decided to remain this sentence after our discussion.
Generally speaking, there were didexist some inadequacies in our work, the reviewers put forward many constructive opinions on our paper and we have made some revises correspondingly. We hope the reviewers will agree with our explanations and revisions and accept this article! If there have some points needed to further modify, please point out and we will make detailed verifications.
Thank you for your detailed reviews again!
Reviewer 4 Report
This manuscript presents the results of investigation of microstructure and mechanical properties of Al alloy 7075 under one-step and double ageing conditions.
I suggest inserting some photos of samples and equipment used. Moreover, I would like to suggest to the authors to explain in detail the number of samples used for their valuable investigation. Did they have replications (more samples) or only repeated measurements on a particular sample, which of course depends on the testing)
Please use „Figure 2“ instead of „Fig. 2“, etc.
Figure 2d – It is DA treatment (please remove “one-step”)
Figure 2, the red line should be mentioned (what represents)
It would be good to mention the etchant for the optical light microscopy.
Figure 3 should have explanation of a) and b) parts.
When peaks and valleys were explaining (Figure 3) regarding the strength, GP zones and η’ phase were mentioned. What is with the formation of the equilibrium phase η mentioned in the introduction?
Please, do not stick the numbers and measurement units (0h, 8h etc.).
Which load was applied for microhardness measurement, and how the measurement was performed (line measurement or …) to obtain average value.
UTS (ultimate tensile strength) abbreviation should be explained.
Please, avoid double appearing of similar sentences (“Fig. 4 shows the tensile fracture morphology of the 7075 Al sheets after different ageing conditions.”; “Fig. 4 shows the SEM fracture morphologies of the samples subjected to different ageing conditions.“).
The following two sentences could be a little bit closer: “Moreover, the plasticity of the samples 168 also decreased after the DA treatment [12]. From Table 2, it can be observed that the ductility of the DA-treated 173 samples was lower than that of the one-step aged samples.”
Please, correct the following: axis:(a) – Figures 5 and 6
„As shown in Fig. 2 and Section 3.1, the average length and width of the grains after one-step ageing are 281.1 and 31.7 μm, and the length and width of the grains in the DA treatment sample are approximately 237.2 and 28.6 μm, respectively. With these parameters, the strengthening effect of grains size was found to be less than 10 MPa. The calculation was carried out according to equation (4), as discussed later.“ - It is unusual to cite some expression or Figure from the next chapter. Please, modify this paragraph.
1.2 μm instead of 1.2 um
0.4 μm instead of 0.4 um
Regarding the expression 1, I would rather state that P represents the probability that some precipitate diameter Dx will be between D1 (min) and D2 (max) and it amounts to 95% (with 5% of probability of type I error, i.e. 100 – 5 = 95%).
„…8.72 nm. Generally,…“ instead of „…8.72 nm, Generally,…“ (period instead of comma).
Table 3. „DA treatment (3 h at 120 °C and 3 h at 160°C)“ instead of „DAtreatment (3 h at 120 °C and 3h at 160°C)“
Table 3. „ageing (16 h)“ instead of „ageing(16h)“
Table 3. „strengthening, ΔσHP“ instead of „strengthening,ΔσHP“
Table 3. „strengthening, Δσd“ instead of „strengthening,Δσd“
For Table 3, at least one calculated number should be proven.
Formulas should be written more carefully.
Table 3. should be replaced after all the expressions (when the indexes and exponents are not variables, they should be written in normal, not italic, etc.). Accordingly, only one sentence should be written, e.g. “Table 3 shows…”
Lines 273, 274 – How the values of yield strength in brackets were obtained? Is this the difference?
Finally, the authors of the present manuscript compared their investigation with the investigation of Emani et al. 2009. Emani et al. applied shorter time for double aging and a little bit higher temperature (for phase 2). The similar properties were obtained in these two investigations. Is the procedure in the present manuscript more expensive (because of longer times)? Accordingly, justification and purpose of the present research should be highlighted more strongly and the scientific contribution should be emphasized.
Author Response
We thank the reviewers for their patient and constructive comments and our respond to their specific points below.
Comment 1.I suggest inserting some photos of samples and equipment used. Moreover, I would like to suggest to the authors to explain in detail the number of samples used for their valuable investigation. Did they have replications (more samples) or only repeated measurements on a particular sample, which of course depends on the testing)
Reply:Thanks for your conscientious comment. We picked the Instron 8802 electro-hydraulic servo testing instrument and the JEOL-2000F field transmission electron microscope (TEM). For the samples, considering that the samples have cut for SEM and TEM analyses after tensile failure, so we didn’t list the photos of samples. What’s more, the photos were explained only in this response letter and didn’t appear in revised manuscript. We have done repeated test (more samples)because these experiments were part of our research.
Figure 1. The equipment adopt in this study (a) the Instron 8802 electro-hydraulic servo testing instrument (b) the JEOL-2000F field transmission electron microscope.
Comment 2.Please use „Figure 2“ instead of „Fig. 2“, etc.
Figure 2d – It is DA treatment (please remove “one-step”)
Figure 2, the red line should be mentioned (what represents)
It would be good to mention the etchant for the optical light microscopy.
Figure 3 should have explanation of a) and b) parts.
Please, do not stick the numbers and measurement units (0h, 8h etc.).
UTS (ultimate tensile strength) abbreviation should be explained.
1.2 μm instead of 1.2 um
0.4 μm instead of 0.4 um
„…8.72 nm. Generally,…“ instead of „…8.72 nm, Generally,…“ (period instead of comma).
Table 3. „DA treatment (3 h at 120 °C and 3 h at 160°C)“ instead of „DAtreatment (3 h at 120 °C and 3h at 160°C)“
Table 3. „ageing (16 h)“ instead of „ageing(16h)“
Table 3. „strengthening, ΔσHP“ instead of „strengthening,ΔσHP“
Table 3. „strengthening, Δσd“ instead of „strengthening,Δσd“
Table 3. should be replaced after all the expressions (when the indexes and exponents are not variables, they should be written in normal, not italic, etc.). Accordingly, only one sentence should be written, e.g. “Table 3 shows…”
For Table 3, at least one calculated number should be proven.
Formulas should be written more carefully.
Reply:We have read the comments of Reviewer’s.What moved us was that the reviewer not only pointed out many points that need to be modified, but also gave specific guidance.We have done corresponding modifications seen in revised manuscript.
Comment 3.When peaks and valleys were explaining (Figure 3) regarding the strength, GP zones and η’ phase were mentioned. What is with the formation of the equilibrium phase η mentioned in the introduction?
Reply: Equilibrium η phase usually evolved in long time ageing or higher ageing temperature, i.e. generated at over-ageing treatment. In our paper, the ageing process didn’t reach over-ageing treatment, so we didn’t pay attention in equilibrium η phase, except for in Part of Introduction.
Comment 4.Please, avoid double appearing of similar sentences (“Fig. 4 shows the tensile fracture morphology of the 7075 Al sheets after different ageing conditions.”; “Fig. 4 shows the SEM fracture morphologies of the samples subjected to different ageing conditions.“).
The following two sentences could be a little bit closer: “Moreover, the plasticity of the samples 168 also decreased after the DA treatment [12]. From Table 2, it can be observed that the ductility of the DA-treated 173 samples was lower than that of the one-step aged samples.”
Reply: Thanks for your conscientious comment. We realized this problem and had delete the sentence of ‘Figure 4 shows the SEM …… to different ageing conditions’ and ‘From Table 2, it can be observed …… the one-step aged samples.’
Comment 5.Please, correct the following: axis:(a) – Figures 5 and 6
Reply: Thanks for your conscientious comment. We haveimproved this presentation accordingly.
Comment 6.„As shown in Fig. 2 and Section 3.1, the average length and width of the grains after one-step ageing are 281.1 and 31.7 μm, and the length and width of the grains in the DA treatment sample are approximately 237.2 and 28.6 μm, respectively. With these parameters, the strengthening effect of grains size was found to be less than 10 MPa. The calculation was carried out according to equation (4), as discussed later.“ - It is unusual to cite some expression or Figure from the next chapter. Please, modify this paragraph.
Reply: Thanks for your conscientious comment. According your suggestion, we have modified the expression about this part and could be seen in revised manuscript.
Comment 7. Lines 273, 274 – How the values of yield strength in brackets were obtained? Is this the difference?
Reply: Thanks for your conscientious comment. The values, which listed inbrackets were obtained by the experimental data subtracted the calculated data, i.e. 447.5 – 401.6=45.9 MPa. We also amended the expression about this point.
Comment 8.Finally, the authors of the present manuscript compared their investigation with the investigation of Emani et al. 2009. Emani et al. applied shorter time for double aging and a little bit higher temperature (for phase 2). The similar properties were obtained in these two investigations. Is the procedure in the present manuscript more expensive (because of longer times)? Accordingly, justification and purpose of the present research should be highlighted more strongly and the scientific contribution should be emphasized.
Reply: Thanks for your conscientious comment. The procedure did spend more time but the ageing temperature was lower than that paper. At the end of Part of Discussion, we emphasized the value of our research.
Generally speaking, there were didexist some inadequacies in our work, the reviewers put forward many constructive opinions on our paper and we have made some revises correspondingly. We hope the reviewers will agree with our explanations and revisions and accept this article! If there have some points needed to further modify, please point out and we will make detailed verifications.
Thank you for your detailed reviews again!

Round 2
Reviewer 1 Report
The authors made miniscule changes to the manuscript, though nothing which would comply with the term “major revision”. At least the description of the experiments has improved. To my surprise, the simulation results shown in Table 3 have changed a lot, although the input parameters are identical -- very strange indeed!
Author Response
We thank the reviewer for the rigorous comments.Ourresponseto the specific points are given below.
Comment 1. The authors made miniscule changes to the manuscript, though nothing which would comply with the term “major revision”. At least the description of the experiments has improved. To my surprise, the simulation results shown in Table 3 have changed a lot, although the input parameters are identical -- very strange indeed!
Reply: Thank youfor your comment.We realized the problem with ”the contribution of precipitates was smaller than other strengthening mechanisms,” in the original manuscriptand verifiedthe volume fractions of the precipitates(the volume fractions were verified to be 5.25±0.3% and 6.57±0.3% for the one-step ageing and double ageing processes, respectively) according to your suggestion and corresponding literature before the submission to Materials. However, we forgot to update the data in Table 3 due to our negligence, and rather did the verification directly in the Review Report (Round 1).We hereby explain our verification and beg for your understanding.
Truthfully, we have learned from your rigorous academic approachand will do better in future research.Thank you again for your detailed review!
Reviewer 2 Report
Thank you very much for the careful revision job!
The final revision of the linguage is desirable.
Author Response
Comment 1.The final revision of the linguage is desirable.
Reply:Thank you for yourencouragement and approval of our work!. The manuscript has been submitted to an editing service to revise and improve the language.
Reviewer 3 Report
The Authors have made many changes to the article. However, there are some points that need to be corrected. The text on line 257-271 is not in log format. Need to fix: font size and style. This is not a critical remark, it is possible to correct it by the editor. The scientific side has undergone significant improvements. The article can be accepted in present form.
Author Response
Comment 1.The scientific side has undergone significant improvements.
Reply:Thank you for your encouragement! We continually strive toenhance our research capabilities and develop a rigorous scientific research approach.
Reviewer 4 Report
The authors did an effort to make the improvements. However, some very important suggestions are not taken into the consideration. Below are my comments from Review 1 which, I think, were not taken into account and consequently, I wrote additional explanations (in bold).
- I suggest inserting some photos of samples and equipment used. Moreover, I would like to suggest to the authors to explain in detail the number of samples used for their valuable investigation. Did they have replications (more samples) or only repeated measurements on a particular sample, which of course depends on the testing).
When some research is conducted, the number of samples should be known and pictures should be assured. Furthermore, the question “Did they have replications (more samples) or only repeated measurements on a particular sample, which of course depends on the testing)” is not answered.
- Please use „Figure 2“ instead of „Fig. 2“, etc. - not Figure. 2
- Please, do not stick the numbers and measurement units.
- Regarding the expression 1, I would rather state that P represents the probability that some precipitate diameter Dx will be between D1 (min) and D2 (max) and it amounts to 95% (with 5% of probability of type I error, i.e. 100 – 5 = 95%).
Unfortunately, the authors have written the suggested word type as tape, which could mean that they probably do not understand the topic.
5. For Table 3, at least one calculated number should be proven.
6. Formulas should be written more carefully.
Very important: A newly added formulas (in red) are not written mathematically correct.
- Finally, the authors of the present manuscript compared their investigation with the investigation of Emani et al. 2009. Emani et al. applied shorter time for double aging and a little bit higher temperature (for phase 2). The similar properties were obtained in these two investigations. Is the procedure in the present manuscript more expensive (because of longer times)? Accordingly, justification and purpose of the present research should be highlighted more strongly and the scientific contribution should be emphasized.
If the authors do not ensure this (justification and purpose of the present research should be highlighted more strongly and the scientific contribution should be emphasized), it would mean that they have repeated the research that has already been done. Accordingly, please write in more detail the differences between your research and the mentioned research of Emani et al.
Author Response
Thank you for yourpatient and rigorous comments. Please findour responseto eachspecific commentbelow.
Comment 1.I suggest inserting some photos of samples and equipment used. Moreover, I would like to suggest to the authors to explain in detail the number of samples used for their valuable investigation. Did they have replications (more samples) or only repeated measurements on a particular sample, which of course depends on the testing).
Reply:Thanks for your conscientious comment. We have added the details of the samples and equipment as suggested, shown in Figure 1. Furthermore, Figure 1 (a) shows the shape of the sample and the details regarding the length of the sample, whichis shown in the Supplementary Information (named equipment). Figure 1 (b) illustrates the partial residual materials (from the same batch) after cutting to produce thetensile samples. After cutting, the samples were mixed together and randomly divided into seven groups (every group contained 3 samples), followed by different heat treatments and corresponding tensile tests.During the testing process, to decrease the experimental error, if two results did not fall within the accepted experimental error, two moresamples were tested.
Figure 1.Images of the samples and some equipment usedin this study. (a)Theshape and size ofthe sample, (b)the partial residueof the materials after cutting,(c)a partial image of the Instron 8802electro-hydraulic servo testing instrument, (d) the Shimadzu HMV-G21ST Vickers micro-hardness tester, (e) the Nova Nano-SEM230 scanning electron microscope (SEM), and (f) the JEOL-2000F field transmission electron microscope (TEM) used.
Comment 2.Please use “Figure 2” instead of “Fig. 2”, etc. - not Figure. 2
Reply:Thanks for your conscientious comment. We apologize forignoring this detail. We have done corresponding modifications throughout the manuscript,which can be seen in the revised manuscript.
Comment 3.Please, do not stick the numbers and measurement units.
Reply:We haveinserteda space between the numbers and units throughout the manuscript.
Comment 4.Regarding the expression 1, I would rather state that P represents the probability that some precipitate diameter Dx will be between D1 (min) and D2 (max) and it amounts to 95% (with 5% of probability of type I error, i.e. 100 – 5 = 95%).Unfortunately, the authors have written the suggested word type as tape, which could mean that they probably do not understand the topic.
Reply: Thank you for the suggestion. The corresponding change has been made on lines 282 to 284 in the manuscript. We apologizefor writing “tape” instead of “type.” This was a typographical error and does not mean that we do not understand the topic.
Comment 5.For Table 3, at least one calculated number should be proven.
Reply: We haveimproved this presentation and have added the specific expression for Dislocation strengthening of the one-step ageing treatmentaccordingly. This has been added on line 319 to 323.
Comment 6.Formulas should be written more carefully.Very important: A newly added formulas (in red) are not written mathematically correct.
Reply: We have modified the expression ofVplateto be more acceptable and added some descriptions for the formula.This is shown in the revised manuscriptbetween lines 294 and 295.
Comment 7.Finally, the authors of the present manuscript compared their investigation with the investigation of Emani et al. 2009. Emani et al. applied shorter time for double aging and a little bit higher temperature (for phase 2). The similar properties were obtained in these two investigations. Is the procedure in the present manuscript more expensive (because of longer times)? Accordingly, justification and purpose of the present research should be highlighted more strongly and the scientific contribution should be emphasized.
Reply: Thank youfor your rigorous academic approachand conscientious comment. We have replenished the justification and purpose of the present research, lines 348 to 363,based on your suggestion.The aim of adding thediscussion containing the comparisonwith previous work is to reveal the diversity of the DA treatment and to test our procedure’s practicality. Additionally, our paper includes a discussion onthe various strengthening mechanics accomplished, which is not mentioned in the research of Emani et al.
Generally speaking, some inadequacieswere present in our work, however, these wereearnestly amended according to your many constructivecomments. We hope the reviewers will agree with our explanations and revisions and accept this article. If there are any further modification required, please point these out and we will be happy to make the corresponding detailed revisions.We strive tocontinually enhance our research capabilities and develop a rigorous scientific research approach.
Thank you again for your detailed reviews.
